# Metabolic Adverse Effects of Psychotropic Drug Therapy: A Systematic Review

**Lizeth Sepúlveda-Lizcano** [1]**, Vivian Vanessa Arenas-Villamizar** [1]**, Enna Beatriz Jaimes-Duarte** [2]**, Henry García-Pacheco** [3,4]**, Carlos Silva Paredes** [4,5]**, Valmore Bermúdez** [6] **and Diego Rivera-Porras** [1,*]

[1] Facultad de Ciencias Jurídicas y Sociales, Universidad Simón Bolívar, Cúcuta 540001, Colombia
[2] Facultad de Salud, Universidad de Pamplona, Pamplona 543050, Colombia
[3] Facultad de Medicina, Departamento de Cirugía, Universidad del Zulia, Hospital General del Sur «Dr. Pedro Iturbe», Maracaibo 4002, Venezuela
[4] Facultad de Medicina, Escuela de Medicina, Cátedra de Fisiología, Universidad del Zulia, Maracaibo 4002, Venezuela
[5] Unidad de Cirugía para Obesidad y Metabolismo (UCOM), Maracaibo 4002, Venezuela
[6] Facultad de Ciencias de la Salud, Universidad Simón Bolívar, Barranquilla 080001, Colombia
[*] Correspondence: diego.rivera@unisimon.edu.co

**Abstract:** This review aimed to investigate the metabolic alterations associated with psychopharmacological treatment of neuropsychiatric disorders, which can significantly impact patients' physical health and overall quality of life. The study utilized the PRISMA methodology and included cross-sectional, retrospective studies, and randomized clinical trials from reputable databases like SCOPUS, CLARIVATE, SCIENCE DIRECT, and PUBMED. Out of the 64 selected studies, various psychotropic drug classes were analyzed, including antidepressants, anticonvulsants, and antipsychotics. Among the antidepressants, such as amitriptyline, Imipramine, and clomipramine, weight gain, constipation, and cardiovascular effects were the most commonly reported metabolic adverse effects. SSRI antidepressants like Fluoxetine, Sertraline, Citalopram, Escitalopram, and Paroxetine exhibited a high prevalence of gastrointestinal and cardiac alterations. Regarding anticonvulsants, valproic acid and Fosphenytoin were associated with adverse reactions such as weight gain and disturbances in appetite and sleep patterns. As for antipsychotics, drugs like Clozapine, Olanzapine, and Risperidone were linked to weight gain, diabetes, and deterioration of the lipid profile. The findings of this review emphasize the importance of continuous monitoring for adverse effects, particularly considering that the metabolic changes caused by psychopharmacological medications may vary depending on the age of the patients. Future research should focus on conducting field studies to further expand knowledge on the metabolic effects of other commonly prescribed psychotropic drugs. Overall, the study highlights the significance of understanding and managing metabolic alterations induced by psychopharmacological treatment to enhance patient care and well-being.

**Keywords:** metabolism; psychopharmacological treatment; antidepressants; anticonvulsants; antipsychotics

## 1. Introduction

Metabolic disorders encompass a complex interplay of physiological, biochemical, and clinical factors, primarily characterized by insulin resistance, hyperglycemia, visceral fat accumulation, dyslipidemias, endothelial dysfunction, and elevated blood pressure [1]. Notably, psychiatric conditions like major depressive disorder (MDD), bipolar disorder (BD), schizophrenia, anxiety disorder, attention deficit hyperactivity disorder (ADHD), and post-traumatic stress disorder (PTSD) are associated with an increased risk of various endocrine disruptions [2,3].

Obesity and the accumulation of visceral fat have significant implications for insulin resistance, dyslipidemia, metabolic syndrome, and the risk of type 2 diabetes (T2DM),

non-alcoholic fatty liver disease (NAFLD), cardiovascular disease, cancer, and even mortality [4,5]. Numerous commonly used medications, including certain antidiabetic drugs and neuropsychotropic medications like atypical antipsychotics, antidepressants, and antiepileptics, can lead to weight gain and fat redistribution, particularly the accumulation of visceral fat [6]. However, the underlying mechanisms governing body weight and fat distribution remain poorly understood, hindering the identification of high-risk patients for prevention and the development of safer drugs and targeted treatments to mitigate the risk of cardiometabolic diseases. Given the heightened vulnerability of psychiatric patients to metabolic syndrome (MetS) and its adverse consequences on physical health, urgent attention is required in research, prevention strategies, strict monitoring, and treatment to minimize future MetS in this population.

Among the drugs commonly employed to treat psychiatric disorders, three major classes are associated with cardiometabolic side effects: antidepressants, mood stabilizers, and antipsychotics. Antidepressants are vital for managing anxiety and depressive disorders, while mood stabilizers, such as lithium, valproate (Divalproex), carbamazepine, and lamotrigine, are pivotal in treating bipolar disorder and related mood disorders. Antipsychotics, divided into first-generation (FGAs) and second-generation (SGAs) categories, include well-known agents like fluphenazine, haloperidol, chlorpromazine, aripiprazole, olanzapine, clozapine, quetiapine, and risperidone. Despite their effectiveness in various conditions, SGAs are associated with a notable risk of metabolic complications [7–9].

In this systematic review, we aim to consolidate the latest data concerning the impact of psychotropic drugs on metabolic biomarkers, encompassing body weight, fat distribution, fasting glucose levels, and the diagnosis of type 2 diabetes and dyslipidemias. Moreover, we will discuss recent findings focusing on predictive factors that could identify patients at risk for weight gain and metabolic complications. By shedding light on these crucial aspects, we hope to facilitate better-informed treatment decisions and enhanced patient care for individuals facing these challenges.

## 2. Materials and Methods

This systematic review was carried out following the parameters established by the PRISMA methodology [10] (Table 1). For this, different search equations were designed based on different descriptors, making it possible to delimit and locate the desired information [10]. The research question was formulated using the PICO strategy and was structured as follows: What are the metabolic alterations or changes secondary to psychopharmacological treatment?

**Table 1.** Research question.

|   | Component | Descriptor |
|---|---|---|
| P | Patient/Problem of interest | Patients |
| I | Intervention | Psychopharmacological treatment |
| C | Comparison | N/A |
| O | Outcome | Metabolic alterations and/or changes |

### 2.1. Characteristics of the Studies
Selected Types of Studies

Cross-sectional and retrospective observational studies, as well as randomized clinical trials, were considered.

### 2.2. Participants

The studies selected for this review included information reported from patients with mental health disorders (psychosis, schizophrenia, depression, anxiety, epilepsy, consumption of stimulant drugs, among others) who were under psychopharmacological treatment.

### 2.2.1. Types of Intervention

Those papers that studied the following families of psychotropic drugs were taken into account: tricyclic anti-depressants, SSRI anti-depressants, anticonvulsants, antiepileptics, and anti-psychotics.

### 2.2.2. Types of Performance Measurement

Information was collected related to metabolic changes such as increased body mass index (BMI), a new diagnosis of abdominal obesity, hypertension, diabetes, lipid alterations, glycemic changes, and alterations in cholesterol, triacylglycerides, and HDL cholesterol, and LDL cholesterol.

### 2.3. Sources of Information

It was considered relevant to identify the terms related to the main groups of psychopharmacological treatments and their derivations, the processes, disorders, and/or metabolic alterations that may occur, and their different denominations.

Generating as basic criteria the relevance of the study, the authenticity of the information, quality of the search sources, purpose, topic of interest, existing information on the topic, and central search terms, delimiting the problem of interest, types of psychopharmacological treatment, and metabolic changes or alterations, based on the PICO method.

Once the search terms were recognized, we consulted the Descriptor in Health Sciences (DECS) and the Medical Subject Heading (MESH) to find different synonyms or terms and generate a broader coverage of the variables studied. Table 2 presents the descriptors and related terms.

**Table 2.** DESCH and MESH descriptors.

| MESH and DECS | | |
|---|---|---|
| Patient | Anticonvulsant, Antidepressants, Antiepileptics, Antipsychotic agent, Antipsychotic effect, Antipsychotics, Central nervous system depressants, Dopamine antagonist, GABA modulator, Hypnotic effect, Major Tranquilizers, Neuroleptics, Psychoactive agent, Psychoactive drug, Psychopharmacological treatment, Psychotropic drug, Second generation antidepressants, Sedative effect, Sedatives, Tranquilizers, Tranquilizing agents, Tricyclic antidepressants. | Abdominal Obesity, Abnormal metabolism, Blood pressure, LDL cholesterol, Metabolic, Metabolic disorders, Metabolic Syndrome, Metabolic Syndrome X, Morbid Obesity, Obesity, Pediatric Obesity, Reaven's Syndrome, Severe Obesity, Triglycerides, Visceral Obesity. |

(Source: Information obtained from DECS and MESH).

### 2.4. Search Strategy

The team constructed the search equations in English using the terms found in the DECS and MESH (see Table 3). These were built using key connectors, which act as logical operators. Those used were AND/OR/NOT, and symbols such as " " and ().

Based on the equations, a search was performed in the databases PLOS ONE, SCOPUS, PUBMED, SCIENCE DIRECT, and CLARIVATE.

### 2.5. Selection and Analysis

The selection and analysis process carried out by the authors began in 2022. In order to achieve the search and systematization of the information, the tasks were distributed in 2 work teams. Both teams were in charge of reviewing and recognizing the eligibility of the studies by reading the title, abstract, methodology, and results. The researchers eliminated the articles that did not meet the inclusion criteria and verified if there was a lack or doubt as to the eligibility of the studies, to verify the documents as a team later. Following this, the selected documents were systematized in an Excel file in which the PICO question, search equations, database results, methodology, equations, references, and findings were recorded.

**Table 3.** Search equations.

| Data Base | Search Algorithm |
|---|---|
| CLARIVATE PLOS ONE PUBMED SCIENCE DIRECT SCOPUS | ("Psychopharmaceuticals" OR "Agents Psychoactive" OR "Drugs Psychoactive" OR "Drugs Psychotropic" OR "Psychoactive Agents" OR "Psychoactive Drugs" OR "Psychopharmaceuticals" OR "Psychotropic Drugs" OR "Nootropic Agents" OR "Psychopharmacology") AND ("Obesity" OR "Morbid Obesities" OR "Morbid Obesity" OR "Obesities Morbid" OR "Obesities Severe" OR "Obesity Severe" OR "Severe Obesities" OR "Severe Obesity") |
| | ("Psychopharmacology") AND ("Metabolism") |
| | ("Metabolic alteratios" OR "Metabolic changes" OR "Metabolic Sindrome") AND ("Psychopharmacological treatment" OR "Psycofarmaceutical") |
| | ("Psycoactive agent") AND ("Metabolic Sindrome") |
| | ("Metabolic alteration"AND "psychopharmacological treatment") |
| | ("Psychopharmaceuticals" OR "Agents Psychoactive" OR "Drugs Psychoactive" OR "Drugs Psychotropic" OR "Psychoactive Agents" OR "Psychoactive Drugs" OR "Psychopharmaceuticals" OR "Psychotropic Drugs" OR "Nootropic Agents" OR "Psychopharmacology") AND ("Patient Metabolic" OR "Adolescent Sindrome reaven" OR "Child Obesity" OR "Childhood Obesity" OR "Childhood Onset Obesity" OR "Childhood Overweight" OR "Infant Obesity" OR "Infant Overweight" OR "Infantile Obesity") |
| | ("Psychopharmaceuticals" OR "Agents Psychoactive" OR "Drugs Psychoactive" OR "Drugs Psychotropic" OR "Psychoactive Agents" OR "Psychoactive Drugs" OR "Psychopharmaceuticals" OR "Psychotropic Drugs" OR "Nootropic Agents" OR "Psychopharmacology") AND ("Obesity Infant" OR "Obesity" OR "Obesity Pediatric" OR "Overweight Adolescent" OR "Overweight Childhood" OR "Overweight Infant") |
| | ("Metabolic change") AND ("Psycofarmacological treatment") |
| | ("Psychopharmaceuticals" OR "Agents Psychoactive" OR "Drugs Psychoactive" OR "Drugs Psychotropic" OR "Psychoactive Agents" OR "Psychoactive Drugs" OR "Psychopharmaceuticals" OR "Psychotropic Drugs" OR "Nootropic Agents" OR "Psychopharmacology") AND ("Obesity in Adolescence" OR "Obesity in Childhood" OR "Obesity, Adolescent" OR "Obesity, Child" OR "Obesity, Childhood" OR "Obesity, Childhood Onset") |
| | ("Psychopharmaceuticals" OR "Agents Psychoactive" OR "Drugs Psychoactive" OK "Drugs Psychotropic" OR "Psychoactive Agents" OR "Psychoactive Drugs" OR "Psychopharmaceuticals" OR "Psychotropic Drugs" OR "Nootropic Agents" OR "Psychopharmacology") AND ("Adolescent Obesity" OR "Adolescent Overweight" OR "Child Obesity" OR "Childhood Obesity" OR "Childhood Onset Obesity" OR "Childhood Overweight" OR "Infant Obesity" OR "Infant Overweight" OR "Infantile Obesity") |
| | ("Psicoactive agent" AND "metabolic challeng") |
| | ("Antidepressants" OR "Antidepressant" OR "Tricyclic antidepressants" OR "Tricyclic antidepressant" OR "Second generation antidepressant" OR "Second generation antidepressant") AND ("Metabolic changes") OR ("Basal metabolism") OR ("Metabolic syndrome") OR ("Metabolic syndrome x") OR ("Carbohydrate metabolism") OR ("Abnormal metabolism") OR ("Lipid metabolism") OR ("Triglycerides") OR ("Blood pressure") OR ("Obesity") OR ("Morbid Obesity") OR ("Severe obesity") OR ("Abdominal obesities") |
| | ("Antipsychotic OR "Antipsychotic Agents" OR "Antipsychotic Agent" OR "Antipsychotic Effect" OR "Antipsychotic Effect" OR " Antipsychotic Drugs" OR "Antipsychotic Drugs" OR " Neuroleptics" OR " Neuroleptic" OR " Major Tranquilizers" OR " Tranquilizer" OR "Dopamine antagonist") OR ("Dopamine antagonist") AND ("Metabolic changes") OR ("Basal metabolism") OR ("Metabolic syndrome") OR ("Metabolic syndrome x") OR ("Carbohydrate metabolism") OR ("Abnormal metabolism") OR ("Lipid metabolism") OR ("Triglycerides") OR ("Blood pressure") OR ("Obesity") OR ("Morbid Obesity") OR ("Severe obesity") OR ("Abdominal obesities") |
| | ("Anticonvulsant" OR "Anticonvulsant" OR" Antiepileptic Drugs") AND ("Metabolic changes") OR ("Basal metabolism") OR ("Metabolic syndrome") OR ("Metabolic syndrome x") OR ("Carbohydrate metabolism") OR ("Abnormal metabolism") OR ("Lipid metabolism") OR ("Triglycerides") OR ("Blood pressure") OR ("Obesity") OR ("Morbid Obesity") OR ("Severe obesity") OR ("Abdominal obesities") |
| | ("Hypnotic Effect" OR "Hypnotic Effects" OR "Sedative Effect" OR " Sedative Effects" OR " Sedatives" OR " Sedative" OR " GABA Modulators" OR " GABA modulator" OR "Tranquilizing Agents" OR "Central nervous system depressants") AND ("Metabolic changes") OR ("Basal metabolism") OR ("Metabolic syndrome") OR ("Metabolic syndrome x") OR ("Carbohydrate metabolism") OR ("Abnormal metabolism") OR ("Lipid metabolism") OR ("Triglycerides") OR ("Blood pressure") OR ("Obesity") OR ("Morbid Obesity") OR ("Severe obesity") OR ("Abdominal obesities") |

Finally, this review was carried out in 3 phases: identification, selection, and elimination and inclusion of the PRISMA flowchart (Figure 1).

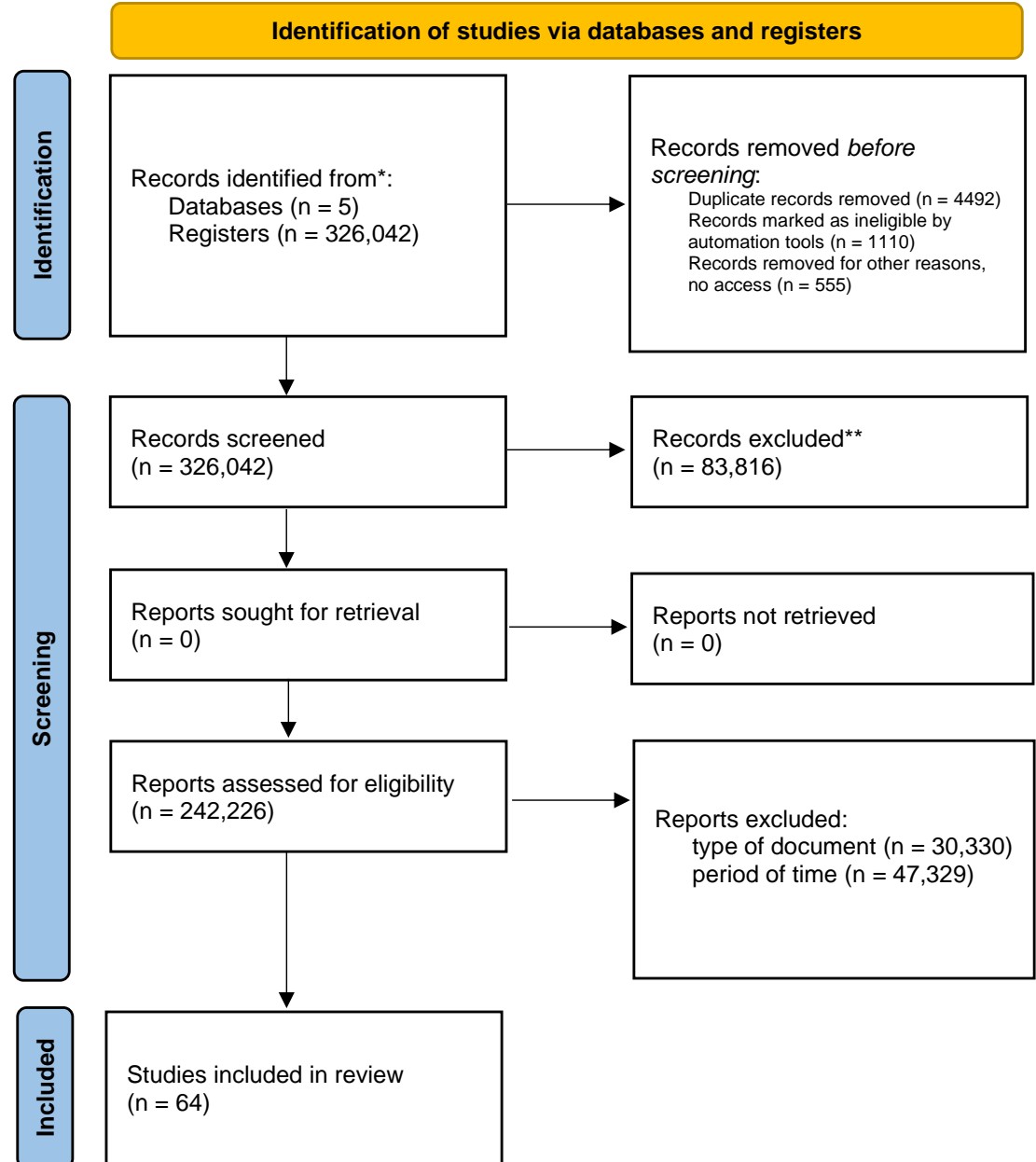

**Figure 1.** Graphical representation of the PRISMA flow * *p* < 0.05, ** *p* < 0.01.

*2.6. Inclusion Phase*

The search was carried out in the databases PLOS ONE, SCOPUS, PUBMED, SCIENCE DIRECT, and CLARIVATE where, as a central strategy of the search, the search equations were run in the databases; 326,042 articles were found, where 83 were discarded. Overall, a further 816 documents were eliminated by type, 30,330 eliminated for non-compliance of criteria by time period, 47,329 documents without access, 555 duplicate documents, 4492 incomplete texts, and 1110 eliminated for non-compliance of variables, leaving as a final result 64 documents (Table 4). Discriminating by database, 22 files remained in PLOS ONE, 12 in SCOPUS, 20 in PUBMED, 5 in SCIENCE DIRECT, and 5 in CLARIVATE (Table 5).

**Table 4.** Filters applied.

| Database | Total Found | Type of Document | Time Period | No Access | Revisions/Incomplete/ Duplicated Texts | Non-Compliance with the Variable Criteria | Total Sample |
|---|---|---|---|---|---|---|---|
| PLOS ONE | 12,317 | 7921 | 2896 | 316 | 575 | 256 | 22 |
| SCOPUS | 3508 | 1561 | 956 | 126 | 249 | 63 | 12 |
| PUBMED | 287,491 | 13,650 | 27,330 | 10 | 2626 | 635 | 20 |
| SCIENCE DIRECT | 21,275 | 6729 | 16,069 | 94 | 1023 | 121 | 5 |
| CLARIVATE | 1451 | 469 | 78 | 9 | 19 | 35 | 5 |
| Total | 326,042 | 30,330 | 47,329 | 555 | 4492 | 1110 | 64 |

**Table 5.** Result discrimination by database.

| Database | PLOS ONE | SCOPUS | PUB MED | SCIENCE DIRECT | CLARIVATE |
|---|---|---|---|---|---|
| Equation (1) | 7 | 4 | 5 | 0 | 2 |
| Equation (2) | 7 | 6 | 4 | 3 | 1 |
| Equation (3) | 3 | 1 | 2 | 2 | 1 |
| Equation (4) | 0 | 0 | 2 | 0 | 0 |
| Equation (5) | 0 | 1 | 2 | 0 | 1 |
| Equation (6) | 0 | 0 | 1 | 0 | 0 |
| Equation (12) | 2 | 0 | 1 | 0 | 0 |
| Equation (13) | 1 | 0 | 1 | 0 | 0 |
| Equation (14) | 1 | 0 | 1 | 0 | 0 |
| Equation (15) | 1 | 0 | 1 | 0 | 0 |
| Total | 22 | 12 | 20 | 5 | 5 |

After eliminating the documents and having a total of 64 selected articles, we proceeded to carefully read the documents and classify the contributions to the search for information, rescuing from them the contributions that answered the established PICO question (Figure 1).

During the process of searching, systematization, registration, selection, and analysis of information, the work team was guided by the ethical parameters for constructing this type of document. It should be noted that this review was developed from primary sources of information. For this reason, no human participants, population samples, or animals were used.

## 3. Results

The tables (Tables 6–8) show the findings regarding psychopharmacological drugs and the alterations or changes that they generate in the metabolic system, showing that diabetes, weight gain, hypertension, constipation, cardiovascular alterations, and alteration of intestinal flora are the most frequently presented in the different families of psychopharmacological drugs. In addition, the alteration in mental health and its relationship with metabolic alterations are highlighted (Table 9) [11,12].

Among the metabolic changes that occur in the treatment with tricyclic anti-depressants (Table 6) such as amitriptyline, Imipramine, and clomipramine, weight gain is very frequent, showing statistical variants between 7% and 19%, in addition to alterations such as constipation which varies between 10 and 12%, cardiovascular side effects, lower blood pressure, alterations in the intestinal flora, nausea, and vomiting. According to statistics, 13.3% of the patients note significant changes in metabolism [11,12].

Studies show that treatments with SSRI anti-depressants where patients are given Fluoxetine, Sertraline, Citalopram, Escitalopram, and Paroxetine present a high prevalence of suffer gastrointestinal alterations ranging from 16.9% to 22.9%, and 12% suffer cardiac alterations. The studies reviewed show a less frequent relationship between risk of pulmonary hypertension in the fetus during gestation, constipation, metabolic disorders, nutritional alteration, postpartum hemorrhage, glycemic alteration in diabetics, and risk of metabolic alteration in patients with coronary disease and digestive alterations.

**Table 6.** Antidepressant metabolic changes.

| Ranking | Medication | Metabolic Change | Effect on the Patient |
|---------|-----------|------------------|----------------------|
| Antidepressants Tricyclics | Amitriptyline | Increased appetite by 19% [13]. Constipation 10% [14]. Weight gain of 10% [15]. Cardiovascular side effects 14% to 17% [14]. | 13.3% of patients with depression or anxiety who receive treatment with these psychotropic drugs for a period longer than 5 months' notice significant changes in metabolism [13,15–17]. |
| | Imipramine | Sedative effects [16]. Constipation 12%. Decreased blood pressure, dizziness 7% [14]. Metabolic risk 12% due to inappropriate drug metabolism [17]. | |
| | Clomipramine | Nausea, vomiting, intestinal flora alliteration and 12% to 17% body weight gain [18]. | |
| Antidepressants (SSRIS) | Fluoxetine | Weight gain (6.8% Cardiac disturbance (12.5) [19]. Pulmonary hypertension in the fetus during gestation (1st to 20%) [20]. | 5.5% 21 to 25.5% of patients who are on selective serotonin reuptake inhibitors have a sedentary lifestyle and non-compliance with diets, which has as a consequence relevant change in their metabolism [20,21]. 38.4% (95% CI: 31.1–45.7) of patients on medication for more than six months report side effects [22]. Effects such as insomnia or hypersomnia due to citalopram consumption decrease in the second week, as long as the consumption of the drug is adequate to the medical order [21]. |
| | Sertraline | Weight gain (7%), 0.8% constipation [22]. Gastrointestinal disorders 0.65% [23]. Metabolic disorders and nutritional alteration 0.45% [24]. | |
| | Citalopram | 4.6% Risk of postpartum hemorrhage [24]. | |
| | Escitalopram | 0.79% glycemic alteration in diabetics [24]. 1.7% risk of metabolic disturbance in patients with coronary heart disease [19]. 1 to 10% risk of metabolic disorders [25]. 16.9% to 22.9% risk of gastrointestinal disturbance [26]. | |
| | Paroxetine | 20–25% of digestive disorders [27]. 10–15% dry mouth and constipation [27,28]. | |

Anticonvulsant drugs (Table 7), valproic acid and Fosphenytoin, generate adverse reactions related in greater prevalence with weight gain, altered appetite and sleep, and increased likelihood of abdominal obesity, it should be noted that psychopharmacological drugs for the treatment of seizures and epilepsy are varied, but research related to the metabolic effects of these drugs is not as extensive [11,12].

**Table 7.** Metabolic Changes observed in Anticonvulsants.

| Ranking | Medication | Metabolic Change | Effect on the Patient |
|---------|-----------|------------------|----------------------|
| Anticonvulsant | Valproic acid | 12.89% Constipation [11,12]. 23.4% Weight gain [11,12]. Appetite and sleep changes [11]. | 100% of patients with diabetes, and hypertension, should not receive treatment with Fosphenytoin. |
| | Fosphenytoin | 79% BMI alteration. 56% increase in abdominal obesity [11,28]. | |

Within the family of anti-psychotic psychotropic drugs (Clozapine, Olanzapine, Risperidone, Quetiapine, Zipracidone, Perphenazine) used for the treatment of disorders such as psychosis, schizophrenia, bipolar disorder, major depressive disorder, and others, frequent adverse reactions are weight gain in between 0.7% and 16% of patients, diabetes in 2.10 to 25%, followed by alterations and/or deterioration of the lipid profile in 19.3%. Reactions of this kind vary according to the psychophysiological characteristics of each organism, alterations in glucose metabolism, hypercholesterolemia, and hypertriglyceridemia in smaller proportions (Table 8) [29–35].

**Table 8.** Anti-psychotic metabolic changes.

| Ranking | Medication | Metabolic Change | Effect on the Patient |
|---|---|---|---|
| Anti-psychotics | Clozapine | 13.8% weight gain [31].<br>2.10% risk of diabetes [30].<br>00.24% deterioration lipid profile [33,34].<br>34% prevalence of metabolic side effects [33].<br>Withdrawal dyskinesias, increased prolactin levels, weight gain, and other metabolic abnormalities [32,33]. | 12.4% of patients reported significant weight gain, reporting an increase of 4.45 kg after 10 weeks of treatment.<br>Patients under 16 years of age are at increased risk of metabolic derangement caused by clozapine use [32,33]. |
| | Olanzapine | 13.8% weight gain.<br>2.10% risk of diabetes.<br>02.4% deterioration of lipid profile [35].<br>12.9% Alteration of glucose metabolism.<br>0.97% Alteration of lipid metabolism. | 12.4% of patients reported significant changes in weight gain, reporting an increase of 4.45 kg after 10 weeks of treatment [35]. |
| | Risperidone | 12.9% alteration of glucose metabolism.<br>07% weight gain [35].<br>25% diabetes [36].<br>19.3% lipid alteration [37]. | 12.4% of patients reported significant changes in weight gain, reporting an increase of 4.45 kg after 10 weeks of treatment [35]. |
| | Zipracidone | Weight gain (7%).<br>34% prevalence of metabolic side effects [33].<br>19.3% lipid alteration [37]. | 30% of the patients present changes in metabolism showing that the weight gain in the controls varies between 2.34 kg and 4.51 kg. |
| | Quetiapine | 16% weight gain [38].<br>19% blood pressure [39].<br>19.3% lipid alteration (varies according to drug dose) [39]. | 16% of the patients presented changes in metabolism showing that the weight gain in the controls varied between 2.34 kg and 4.51 kg after one year and six months of treatment [39,40]. |
| | Zipracidone | 7% weight gain [41].<br>12% blood pressure [41].<br>2.7% alteration of lipid and glycemic parameters [41].<br>Clinically significant metabolic alterations [41,42].<br>23.2% hypercholesterolemia [41].<br>1.7% hypertriglyceridemia [41]. | In 37.6% of patients treated with Zipracidone, the symptoms occur during the first year of treatment [41].<br>19% of the patients at the 12th and 24th week of treatment report changes [41].<br>Tests and major alterations in metabolism [42].<br>3 years after a psychotic episode, patients show an increase in BMI [37,42]. |
| | Perfenazine | 19.3% lipid alteration [37,42]. | |

However, mental health disorders such as depression, mood disorders, schizophrenia, sleep disorders, bipolar behavior disorders, and autism are related to significant metabolic changes, either due to weight gain or loss caused by sedentary lifestyles, or psychophysiological disorders and/or comorbidity with congenital diseases (Table 9) [43].

The information found yields results where there is a comorbidity of obesity, binge eating disorder, and impulsivity and social dependence, a relationship between the endocannabinoid system, temperament, and the development of physical activity in morbidly obese patients [23,44–47], a potential connection between antipsychotic-induced weight gain, cognitive impairment, and gut microbiota [29,48,49], a substantial comorbidity between obesity, mood, and eating disorders, and metabolic syndrome in populations seeking to lose weight [50–52]. Changes in neurotransmitter systems and fatty acid metabolism are associated with a depressive state [14,53,54].

**Table 9.** Alteration in mental health and its relationship with metabolic changes.

| Alterations in Mental Health | Metabolism |
| --- | --- |
| Depression Mood disorders | This is an alteration associated with metabolic syndrome, relating comorbidity with diabetes mellitus type 2 and cardiovascular disorders [43]. Direct relationship between physical and metabolic alterations, the stress regulatory system and the appetite activator, caused by the body's energy deficiency in the major depressive disorder due to imbalance of the hypothalamus-pituitary-adrenal gland, the concentration of hormones, and appetite [33,55]. |
| Schizophrenia Psychotic disorders | Comorbidity of a psychotic disorder with metabolic alterations, hypertension, and cardiovascular risk; within the investigations, the need to follow up on the group of medications that the patient is taking arises [56]. |
| Autism | Decreased levels of oxytocin; this disorder is characterized by the presence of inborn errors in metabolism [12]. |
| Bipolar disorder | The patient presents an elevated glucose metabolism in the amygdala [57]. However, exposure to stressful situations and excessive consumption of sugars and fats produces a high risk of suffering alterations such as obesity, hypertension, and other disorders [58]. |
| Eating behavior disorder | Food is the basis for the development of energy. If the organism presents alterations in eating behavior, whether caused by disorders such as anorexia or bulimia, the processing of enzymes and micronutrients is not optimal [47,58,59]. |
| Sleep disturbances | Just as nutrition is important, rest is vital for the process of growth and protein generation, altering the sleep cycle generates the risk of metabolic alterations [34,60–62]. |

## 4. Discussion

The primary objective of this comprehensive systematic review was to identify and analyze the metabolic changes and alterations that occur in patients undergoing psychopharmacological treatment, specifically focusing on three classes of psychotropic drugs: antidepressants, anticonvulsants, and antipsychotics. The review sought to establish the presence of adverse elements and reactions resulting from the consumption of these drugs. The findings overwhelmingly indicate that 89% of these drugs lead to metabolic reactions such as obesity, morbid obesity, and disturbances in eating behavior, thereby suggesting that hypertension, diabetes, constipation, cardiovascular issues, and alterations in intestinal flora are prevalent adverse effects observed in both the short and long term.

Special attention is warranted for specific groups of patients who may be at higher risk of experiencing such metabolic changes. For instance, children and adolescents under psychotropic medications face a greater likelihood of developing side effects like withdrawal kinesis, increased prolactin levels, weight gain, and other metabolic abnormalities, including type 2 diabetes mellitus. Furthermore, the risk to the adult and older adult populations is compounded by factors such as mental health conditions, medication usage, and sedentary lifestyles [63–65].

These findings underscore the importance of closely monitoring patients who are undergoing psychopharmacological treatment, particularly those who may be susceptible to metabolic changes. For children and adolescents, it is crucial to carefully weigh the potential risks and benefits of these medications. To mitigate the metabolic consequences associated with these drugs, lifestyle interventions should be implemented, promoting physical activity and adopting healthy dietary practices. Additionally, healthcare professionals must remain vigilant in monitoring and managing the metabolic health of adult and older adult patients who are receiving psychotropic medications. Regular health screenings, comprehensive assessments, and personalized interventions can help minimize adverse metabolic effects and improve overall well-being in these vulnerable populations.

As the understanding of psychopharmacology and its impact on metabolism continues to advance, further research is needed to develop targeted interventions and optimize the use of psychotropic drugs while minimizing their potential adverse effects on metabolic health. This knowledge can pave the way for more effective treatment strategies and improved patient outcomes in the field of neuropsychiatric disorders.

Depression has been associated with reduced levels of neurotransmitters like dopamine, serotonin, and norepinephrine in the brain. Consequently, antidepressant drugs are designed to increase the availability of these substances through novel mechanisms of action [23]. However, our research indicates that these medications often lead to alterations in appetite, resulting in reduced motivation for activities and movement, leading to weight gain and an increased risk of conditions like type 2 diabetes, hypertension, intestinal flora imbalances, and cardiovascular disorders. Moreover, the impact of these drugs on metabolism can vary depending on the age of the patients. In 2017, discussions arose concerning the potential cardiac alterations associated with antidepressants and the combination of non-prescription drugs. As a result of this research, adherence to psychopharmacological treatments and close monitoring of patients, along with early reporting of adverse effects, were highlighted as crucial recommendations [66,67].

Epilepsy is a chronic brain disorder characterized by recurrent seizures, and anticonvulsant drugs are commonly prescribed to manage these seizures. However, the use of anticonvulsants must be carefully tailored to the specific needs of each patient, and these drugs are also employed for other seizure-related disorders. Our findings reveal that patients with conditions like epilepsy who undergo psychopharmacological treatments involving anticonvulsants such as valproic acid and Fosphenytoin are at risk of experiencing adverse effects like constipation, appetite and sleep disturbances, BMI disorders, or even obesity. To evaluate cardiovascular risk and diabetes, metabolic syndrome serves as a useful tool, allowing for better patient management and appropriate adjustments in pharmacological treatment processes [48,51,68]. Psychological treatment and behavioral regulation are of paramount importance in managing these conditions. However, it is important to note that the information presented here is only a subset of the various anticonvulsant drugs available, and further research is needed to comprehensively understand their metabolic effects.

In light of the potential metabolic consequences associated with both antidepressants and anticonvulsants, healthcare providers should prioritize continuous monitoring and individualized treatment plans. Close collaboration between psychiatrists, neurologists, and other healthcare professionals can help ensure that patients receive the most suitable medications and necessary support to manage potential side effects effectively.

Antipsychotic drugs play a crucial role in treating severe mental illnesses characterized by symptoms like psychosis, schizophrenia, and bipolar disorder, which may present with delusions, hallucinations, and dementia [69]. Certain antipsychotic psychotropic medications, including Clozapine and newer drugs like Zipracidone, have been associated with weight fluctuations and metabolic changes. As early as 2004, studies demonstrated the impact of psychopharmacotherapy on metabolic and hormonal alterations [70]. Researchers have identified specific genetic variations in the LEP and HTR2C genes that may be linked to psychopharmacological treatment and its influence on BMI in patients with schizophrenia [31,33,34,54]. Another hormone, ghrelin, responsible for regulating food intake and body weight, experiences changes in patients with schizophrenia receiving olanzapine, a drug that is associated with decreased ghrelin levels in the blood, similar to what occurs in obesity [21,71]. Furthermore, psychopharmacological treatment has been found to affect ghrelin and leptin levels in patients with schizophrenia, potentially increasing cardiovascular risk in those with metabolic syndrome [52,72].

A sluggish metabolism can profoundly impact a patient's overall well-being, leading to weight gain, fatigue, anhedonia, and difficulties in accomplishing daily tasks and goals, often contributing to depressive moods and a sense of physical and mental exhaustion. In such cases, proper nutrition becomes crucial to regulate metabolic functions. However, the

combination of drug therapy without a healthy diet, low in fat but with adequate amounts of sugar and protein, can disrupt efficiency and productivity. This scenario may lead to stress, and individuals may resort to self-medication with sleeping pills or anxiolytics, even without proper medical prescriptions [73,74]. Studies have found a connection between metabolic syndrome, obesity, and increased impulsivity, food addiction, and depressive symptoms, particularly in middle-aged and elderly individuals with metabolic syndrome [75].

Considering the complex interplay between psychopharmacological treatments, metabolic changes, and mental well-being, a comprehensive approach to patient care is essential. Healthcare providers should not only focus on medication management but also prioritize dietary and lifestyle interventions to support metabolic health. Additionally, regular monitoring of patients' metabolic parameters, coupled with psychological support, can help mitigate the adverse effects and improve overall treatment outcomes. Further research is needed to better understand the intricate relationships between psychopharmacology, metabolism, and mental health, enabling the development of more personalized and effective treatment strategies for patients with severe mental illnesses.

Research conducted between 2018 and 2023 has revealed a concerning trend: a significant number of psychiatric patients face a heightened risk of developing metabolic syndrome. Various factors, such as gender, marital status, employment status, and support networks, exert a considerable influence on this risk. To effectively address the related consequences, it becomes imperative to identify the root causes contributing to this phenomenon [38]. It has been observed that individuals who experience psychotic symptoms at an early age and receive psychopharmacological treatment while maintaining a healthy diet and seeking an optimal quality of life tend to have a lower risk of developing metabolic syndrome compared to those with sedentary lifestyles who do not fully adhere to their prescribed treatments [22,76,77].

However, it is crucial to recognize that certain antipsychotic medications necessitate strict adherence and careful monitoring due to their strong association with elevated body mass index (BMI), obesity, an increased risk of diabetes, significant deterioration of lipid profiles and blood sugar levels, elevated prolactin levels, and hypertension. Managing the potential side effects on metabolic and cardiovascular health becomes paramount when administering these drugs [22,76,77].

To address the concerning prevalence of metabolic syndrome in psychiatric patients, healthcare providers should adopt a comprehensive approach that combines psychopharmacological interventions with lifestyle modifications. Encouraging patients to maintain a healthy diet, engage in regular physical activity, and adhere to their prescribed treatments can significantly mitigate the risk of metabolic disturbances. Moreover, fostering strong support networks and promoting overall well-being in these patients can play a vital role in improving their metabolic health outcomes.

Finally, as the understanding of the intricate relationship between mental health, psychopharmacology, and metabolic syndrome continues to evolve, ongoing research is crucial to develop personalized treatment plans that prioritize both mental and physical well-being. With a holistic approach to patient care, healthcare professionals can help psychiatric patients improve their overall health and quality of life while effectively managing the potential side effects associated with psychotropic medications.

## 5. Conclusions

Metabolic syndrome (MS) is a set of risk elements related to weight gain and loss and other health problems such as hypertension, diabetes, and cardiac alterations, which develop in human beings when they do not take care actions, such as good nutrition, physical activity, and the promotion of actions for good mental health. Among the factors associated with the probability of developing metabolic syndrome is the comorbidity of this syndrome with mood disorders, behavioral alterations, and the use of some psychopharmacological treatments [77].

The metabolic changes caused by psychopharmacological medication vary according to the age of development of the subjects, psychophysiological and hereditary characteristics, and congenital diseases; for this reason, the monitoring of adverse effects should be carried out constantly. It is pertinent that the interdisciplinary team in charge of assigning treatment to its patient takes into account the sociocultural elements, life cycle, support networks, and personality elements of the patients with psychiatric assignment of anti-depressant, anti-psychotic, and anticonvulsant treatments, since these medications as identified in the search for information are directly related to certain conditions and/or deterioration of the metabolic system, since their compounds generate changes in the enzymes Pyruvic (glucose production) and Carboxylase (breaks down fatty acids and amino acids), and can cause alterations such as diabetes, hypertension, high cholesterol, obesity, a significant increase in BMI, and increased abdominal fat.

Within the families of psychotropic drugs studied, it is evident that tricyclic anti-depressants, SSRI anti-depressants, anticonvulsants, and anti-psychotics, of which it is recognized that the effects on the metabolic system occur more frequently in the family of anti-depressants, where the increase in body weight, cardiac alterations, diabetes, gastrointestinal disorders, cardiovascular alterations, constipation and alterations in the intestinal flora, gastric alterations, risk of postpartum hemorrhages, and pulmonary hypertension in pregnant women are only some of the most frequent adverse effects of treatment.

On the other hand, it is pertinent to mention that antipsychotic drugs, which require greater rigor and control in taking the medication, are the ones most related to increased BMI, obesity, risk of diabetes, significant deterioration of the lipid profile and glycemia, increased prolactin levels, and hypertension.

Finally, anticonvulsant psychotropic drugs generate alterations such as changes in the sleep cycle, weight gain, and frequent changes in BMI, caused by an increase in the patient's appetite and drowsiness. It is pertinent to recognize that this family of psychotropic drugs is the one that presents the least information in current studies, and for this reason, it is pertinent to broaden the field of research in this area.

**Author Contributions:** Conceptualization: L.S.-L., V.V.A.-V., E.B.J.-D., H.G.-P., C.S.P., V.B. and D.R.-P.; Investigation: L.S.-L., H.G.-P., C.S.P., V.B. and D.R.-P.; Methodology: L.S.-L., H.G.-P., C.S.P., V.B. and D.R.-P.; Writing—original draft: L.S.-L., V.V.A.-V., E.B.J.-D., H.G.-P., C.S.P., V.B. and D.R.-P.; Writing—review and editing: L.S.-L., V.V.A.-V., E.B.J.-D., H.G.-P., C.S.P., V.B. and D.R.-P.; Funding acquisition: D.R.-P. and V.B. All authors have read and agreed to the published version of the manuscript.

**Funding:** (1) Ministerio de Ciencia, Tecnología e Innovación-Colombia and La Universidad Simón Bolívar-Colombia Joint Grant for strengthening health science, technology, and innovation for ongoing projects with young talent and regional impact. Call # 874-2020; Grant number (Contrato): No. 462, 2021. (2) Internal funds for research strengthening from Universidad Simón Bolívar, Vicerrectoría de Investigación, Extensión e Innovación, Barranquilla, Colombia.

**Institutional Review Board Statement:** Not applicable.

**Informed Consent Statement:** Not applicable.

**Data Availability Statement:** Not applicable.

**Conflicts of Interest:** The authors declare no conflict of interest.

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
