# Peer review of "Metabolic Adverse Effects of Psychotropic Drug Therapy: A Systematic Review"

_ejihpe, doi:10.3390/ejihpe13080110_

Round 1
Reviewer 1 Report
The review is well performed, up to date and relevant to the clinical practice of psychiatry. The paper is not original, as there are several review of the literature about metabolic syndrome and psychotropic medication, but this one compensate in the lack of originality by the rigorousness of the review, the wide array of medication included and the clarity of the presentation for the results.
Only minor revisions of the language and grammar are needed for the paper to be suited to publication in my opinion. Here it follows:
There is a spelling error for the antipsychotic Risperidone, which is incorrectly spelled in the spanish spelling risperidona with the a at the end instead of the e. This happens only in the table of results. A similar spelling check is needed for the antipsychotic perfenazine, incorrectly spelled perfenazina in the table of results.
In the final discussion a revision of the phrasing is needed, between lines 318 and 321. It is not clear what the paragraph meaning is: is it that antipsychotic were the most impacting on metabolic syndrome? If so, a grammar check is needed and the word "where" is incorrectly spelled.
Reviewer 2 Report
The methodology presents very serious errors that do not agree with the results shown (the same sections twice, it does not include antipsychotic drugs and then appear in results and discussion, it indicates that clozapine is a first generation drug...) The analysis of the results is defective and the discussion is very poor.
I therefore don’t recommend publication.
The detailed reasons for not recommending this article are:
1. In the first place, the review is not original nor does it contribute any new data to what has already been published up to now. What new does this article bring? It does not present an adequate justification.
2. I consider that this article has been done quickly and inaccurately, superficially and not carefully.
- A lack of scientific writing has been observed.
- Line 260-261: They make a huge mistake, indicating that clozapine is a first generation antipsychotic. If you investigate antipsychotics, you should know at least their classification.
- They don't know how to put references. Throughout the text they are misplaced. You cannot put [1][2][3][4], you must indicate [1-4] It is important to follow the appropriate rules regarding references.
- Section 2.6 and 2.7 is the same.
- Section: 2.2.1. Types of intervention: where are the antipsychotic drugs? The entire article talks about them, and they are not included in the intervention.
- The tables are cumbersome, very long and with certain errors in the language.
- The organization of the manuscript needs to be improved. It was difficult to follow the manuscript.
- Genes should be in italics.
3. The discussion is poor and superficial for the apparent search and review.
Reviewer 3 Report
The study of psychotropic drugs is an emerging research field. In this review, the author summarized the metabolic changes generated by psychopharmacological drugs and mental health alterations. In general, the review has value and can attract the interest of readers in this field. Following comments may further improve the manuscript.
Specific comments:
1. The section "Abstract" are suggested to be concise. It is better to describe general ideas rather than specific examples of adverse reactions to anti-depressants.
2. Introduction, the necessity of this review should be enhanced.
3. The logic of the article is so poor that it is hard to read. It is necessary to strengthen the connections between sentences and paragraphs.
4. The writing is verbose in many places, especially the complex sentence description. Please simplify your grammar.
Round 2
Reviewer 2 Report
The paper has been strongly improved and it is ready for publication.
Author Response
Muchas gracias estimado revisor.